# Optical Methods for Optimizing Fluorescence Imaging Field of View and Image Quality in Surgical Guidance Procedures

**DOI:** 10.3390/diagnostics14171996

**Published:** 2024-09-09

**Authors:** Jeongmin Seo, Jina Park, Kicheol Yoon, Sangyun Lee, Minchan Kim, Seung Yeob Ryu, Kwang Gi Kim

**Affiliations:** 1Premedicine Course & Department of Medicine, College of Medicine, Gachon University, 8-13, 3 Dokjom-ro, Namdong-gu, Incheon 21565, Republic of Korea; jungmin00010@naver.com (J.S.); jnpr444@gmail.com (J.P.); kcyoon98@gachon.ac.kr (K.Y.); 2Medical Devices R&D Center, Gachon University Gil Medical Center, 21, 774 Beon-gil, Namdong-daero, Namdong-gu, Incheon 21565, Republic of Korea; kormd98@naver.com (M.K.); wyverns0723@gmail.com (S.Y.R.); 3Department of Radiological Science, Dongnam Health University, 50 Cheoncheon-ro 74 gil Jangan-gu, Suwon 16328, Republic of Korea; leesy2024@dongnam.ac.kr; 4Department of Biohealth & Medical Engineering Major and Biomedical Engineering, Gachon University, 1342 Seongnamdaero, Sujeong-gu, Seongnam 13120, Republic of Korea; 5Department of Health Sciences and Technology, Gachon Advanced Institute for Health Sciences and Technology (GAIHST), Gachon University, 38-13, 3 Dokjom-ro, Namdong-gu, Incheon 21565, Republic of Korea

**Keywords:** LED, beam mirror, linear polarized filter, light reflection removal, fluorescence emission

## Abstract

Cancer surgery is aimed at complete tumor resection and accurate lymph node detection. However, numerous blood vessels are distributed within the tumor, and the colors of the tumor, blood vessels, and lymph nodes are similar, making observations with the naked eye difficult. Therefore, tumors, blood vessels, and lymph nodes can be monitored via color classification using an operating microscope to induce fluorescence emission. However, as the beam width of the LED required to induce fluorescence emission is narrow and the power loss of the beam is significant at a certain working distance, there are limitations to inducing fluorescence emission, and light reflection occurs in the observation image, obstructing the view of the observation area. Therefore, the removal of reflected light is essential to avoid missing the diagnosis of the lesion under observation. This paper proposes the use of a beam mirror and polarizing filter to increase the beam width and beam intensity. The refraction and reflection effects of the beam were utilized using the beam mirror, and the rotation angle of the polarizing filter was adjusted to remove light reflection. Consequently, the minimum beam power using the beam mirror was 10.9 mW, the beam width was doubled to 40.2°, and more than 98% of light reflection was removed at 90° and 270°. With light reflection effectively eliminated, clear observation of lesions is possible. This method is expected to be used effectively in surgical, procedural, and diagnostic departments.

## 1. Introduction

Medical systems and surgical methods are advancing owing to the convergence of modern medicine, science, and technology. Consequently, the treatment outcomes for various diseases are improving [1]. In particular, the early detection of cancer has become possible, unlike in the past, through the widespread use of endoscopy [2]. The early detection of cancer considerably affects treatment methods and prognosis. According to statistics, when cancer is detected early, the average five–year survival rate can increase to 91% [3,4].

Malignant tumors are highly invasive and metastasize rapidly. Among various cancer treatment methods, surgical resection aims to remove locally occurring cancer tissues [5]. However, tumors often have excessively developed blood vessels, and tumors and blood vessels are similar in color, making them difficult to distinguish with the naked eye [6]. Consequently, blood vessels may be damaged during tumor removal, and insufficient tumor tissue may be resected to minimize blood vessel damage, leading to cancer recurrence after surgery [7].

A surgical microscope is used to observe the boundaries between tumors and blood vessels. It is widely used in tumor removal surgery because it provides color images to users. However, distinguishing the boundary between the tumor and blood vessels remains challenging because of their similar colors.

To overcome this limitation, a fluorescent contrast agent can be used in conjunction with a surgical microscope. When a fluorescent contrast agent is injected intravenously to stain the tumor and light−emitting diode (LED) light of wavelength 405 nm or 530 nm is directed at the tumor under a surgical microscope, a chemical reaction is induced in the contrast agent, resulting in fluorescence emission with a wavelength of 530 nm to ~560 nm [7,8]. If the spectral color of the fluorescence emission is detected using a camera sensor and imaged, only tumors expressing fluorescence can be distinguished and observed through external monitoring. However, when an LED is used, the intensity of the beam weakens toward the edge of the illuminated area. Additionally, the larger the area illuminated by an LED, the lower the LED intensity in that area [8]. LEDs are categorized into narrow−beam−width and wide−beam−width types, each with its own advantages and disadvantages. LEDs with a narrow beam width have a relatively high beam intensity, which is more advantageous for fluorescence expression; however, the field of view for observing lesions is narrow. In contrast, LEDs with a wide beam width provide a relatively broad field of view for lesion observation; however, the overall beam intensity is lower, and there is a greater difference in fluorescence intensity or brightness across the field. These drawbacks can hinder accurate diagnostic vision when using an operating microscope and a fluorescent contrast agent.

To address this problem, a quasi-symmetrical beam irradiation method using mirrors was studied. By installing four mirrors under the LED and measuring the light intensity, the intensity was observed to be approximately four times higher than that when no mirrors were installed. Additionally, the beam intensity became uniform across the illuminated area, and the area where the fluorescence could be detected expanded [9]. A method for increasing the beam intensity and beam width by minimizing the number of LEDs using asymmetrical irradiation has also been studied [10].

A previous study has observed that, when a lesion is illuminated with an LED, light reflection can obscure the view of the lesion, hindering accurate diagnosis. In medical diagnosis, the cumbersome task of avoiding light reflection is often managed by adjusting the camera angle according to the situation [10]. There are reports of methods that remove light reflections using software; however, these methods involve the inconvenience of using software post−imaging to remove reflections, which complicates the video editing process. Moreover, obtaining results after reflection removal is time−consuming, and color images may degrade during editing, making it difficult to achieve quick and accurate diagnostic results. Although several studies have addressed these issues, comprehensive studies on this topic are scarce [11].

Another study involved canceling light reflection by irradiating an LED from various angles to equalize the beam of the video image and the reflected beam [12]. However, applying this method in a surgical setting is not feasible because of the complexity of adjusting the beam direction and focal distance of the LED [13]. Meanwhile, a method using deconvolution analysis was explored to detect and remove only the wavelength band associated with light reflection by extracting data on the brightness of the reflection and the color of the lesion image using RGB information [14]. However, RGB information is lost during the reflection removal process, necessitating the restoration of the damaged color area and the preservation of the lesion image through a separate source of color image information. However, this method requires a significant amount of data processing and complex calculations.

Image files must be extracted and pretrained to collect image frames without light reflection through continuous shooting. Additionally, the sum and difference of the frames must be classified by comparing the actual captured frames with the prelearned frames. A car image is produced through this classification process. This is followed by a complex calculation process for generating an image without light reflection, which requires extensive data learning [15]. Another proposed method involves connecting a polarizing filter to a camera and a sensor application technology that automatically detects changes in the image and detects only the RGB values for light reflection. However, applying a polarizing filter can degrade the image quality, darken the image, and reduce the imaging radius owing to potential resistance loss in the filter material, thereby narrowing the observation field for accurate diagnosis [16,17,18,19]. Therefore, it is difficult to obtain accurate diagnostic results.

This paper proposes a method to increase the beam width and beam intensity of the LED used for observing fluorescent specimens and to efficiently reduce the light reflection that may appear on the observation monitor.

## 2. Research Method

### 2.1. Analysis of Fluorescence Emission Phenomenon

The fluorescent contrast agent was injected into the patient’s vein using a syringe (Figure 1a). It stained the tumor, and when the tumor was irradiated with a 405 nm LED, a chemical reaction occurred in the contrast agent within the tumor. This chemical reaction generated fluorescence emission. As the wavelength had a spectral color, the fluorescence emission wavelength was detected using a camera sensor, and the tumor was displayed in color on the monitor. This made them easy to observe with the naked eye. The camera sensor detected the fluorescence wavelength, and the fluorescence appeared on the monitor, as shown in Figure 1b [9].

As shown in Figure 1b, the observation procedure is crucial for checking whether any cancer tissue remains after the surgical procedure. Typically, observation is performed using surgical CT, ultrasound, navigation systems, and surgical microscopes to check the tumor removal status. However, as it is challenging to determine the boundary between the tumor and the surrounding tissue, fluorescent contrast agents are used to distinguish the tumor from the surrounding tissue via color differences. This allows easy observation of the boundary, residual tumor status, and tumor removal status with the naked eye through camera imaging and monitoring.

The preparation of fluorescent contrast agents is crucial for lesion monitoring. A 5 mg/mL dilution was prepared by mixing 500 mg of the fluorescent contrast agent with 100 mL of physiological saline. Then, 0.8 mL of this mixture was used to prepare a vial phantom in a 1 mL microtube. The injection volume was maintained at 0.2 and 0.4 mL [21]. After the fluorescent contrast agent was administered, tumor uptake appeared yellow or light green, and spectral measurements confirmed a wavelength range of 530–560 nm. Figure 1c shows the resection status after tumor resection during open surgery. Following the injection of the fluorescent contrast agent, photographs were taken using a camera, and the LED was prepared for illumination. During this process, the operating room lights were turned off to maintain darkness, and the color representation was observed to monitor the tumor resection status.

There are several types of contrast agents, as shown in Figure 2. The fluorescent contrast agents that cause fluorescence in tumors include yellow dye and 5−ALA, as they adhere well to the components found in large quantities in tumors, such as glucose and albumin. They adhere well to serum proteins and fluoresce in the blood vessels. The fluorescent contrast agent that induces fluorescence is ICG. The fluorescent contrast agent commonly used in clinical settings for tumor mucosa monitoring is a yellow dye with an incident (irradiation) wavelength of 405 nm and a fluorescence emission wavelength band of 530–560 nm owing to chemical reactions [22].

The LED irradiation power (*P_o_*) required to induce fluorescence emission using a surgical microscope must be maintained at a minimum of 0.5 mW from the working distance (WD). The LED power is absorbed by the air existing between the LED and the lesion as a function of the WD [9]. Therefore, as a loss of 58% occurs when the power decreases by 1/r^2^ if a general LED is used, the power (*P_o_*) reaching the target is approximately 8.0 pW, resulting in difficult conditions for fluorescence emission [9,22,23,24].

### 2.2. Analysis of Methods for Increasing Beam Width and Beam Intensity

The irradiation range (*θ*) of the LED beam is approximately 20°, as shown in Figure 3 [9]. As the LED beam irradiation width is narrow compared with the entire lesion area, the beam cannot cover the entire lesion. Therefore, it is necessary to expand the beam irradiation range of the LED and increase the power that can reach the lesion as much as the WD allows.

In a typical LED, as shown in Figure 1a, the LED beam (*P_o_*) is irradiated in a triangular shape from the LED to the target, and the irradiated beam is distributed from −i_2_ to i_2_ from the center of i_0_ [9]. At this time, the power (*P_o_*) of the LED beam is the highest at i_0_, and as it reaches i_2_ (or −i_2_), the power (*P_o_*) of the beam decreases. In other words, the distribution and power of the beams are not uniform. This difference in beam distribution and power affects the fluorescence emission, limiting the area of lesion observation. This results in a narrower field of view for the observer, making it difficult to obtain accurate diagnostic results. To overcome these limitations, it is important to supply sufficient beam irradiation power to enable fluorescence emission, as shown in Figure 3a, and ensure that the beam distribution is spread from −i_n_ to i_n_ when i_0_ is centered, allowing the entire lesion to be observed. To achieve this, total−reflection mirrors (m_1_–m_4_) were installed around the LED, as shown in Figure 3b. The beam (*P_o_*) irradiated from the LED was designed to reach m_1_, and the refraction range (*θ* ≥ 40°) was determined through m_3_ [9]. The range of refraction was determined by the reflection angle (*θ*), and the beam with the determined reflection angle was reflected through m_1_, ensuring uniform power (*P_o_*) and an irradiation range from −i_n_ to i_n_ when i_0_ is the center. This suggests that even large lesions can be irradiated, enabling sufficient fluorescence emission.

### 2.3. Analysis of Light Reflection Reduction Methods

Methods for reducing light reflection generally include the use of ultraviolet (UV) light, skylight, polarizing (PL) filters, and neutral density (ND) filters [25]. A PL filter is used to reduce diffuse reflection, and an ND filter is used to uniformly reduce the amount of light and adjust the overall color balance of the image to effectively reduce light reflection. However, most of these filters are only suitable for shooting in natural environments during the day, which limits their use in clinical settings. Therefore, a polarization control method to reduce light reflection is considered more appropriate for use in clinical settings.

Light reflection is based on Snell’s law, as shown in Equation (1), where the angles of incidence and reflection are equal, as illustrated in Figure 4. When light passes through different media, it can be easily affected by the refractive index (n_a_//n_b_) for each medium. Here, light is refracted at an angle of *θ_t_* when incident at an angle of *θ_i_*, and it is also reflected at an angle of *θ_r_*. These conditions create the potential for light reflection [20,26].
(1)$$\frac{\sin\theta_{i}}{\sin\theta_{t}}=\frac{n_{b}}{n_{a}},\theta_{r}=\theta_{i}$$

The causes of light reflection vary anatomically, as shown in Figure 5. However, light bouncing still occurs because of differences in tissue density and the amount of moisture in the mucous membrane. When light bounces off the mucous membrane, it forms a reflection angle (*θ_r_*) and begins to bounce in all directions owing to the difference in refraction angle (*θ_t_*), depending on the inclination angle of the mucous membrane or the inclination angle of the moisture. The light that does not bounce off is absorbed by the tissue mucosa [27]. Therefore, depending on the refraction angle, light reflection can be classified into three types: specular, diffuse, and scattering, which result in unpredictable light reflection. Specular reflection occurs when light touches a smooth, pore−free surface and is reflected in a specific direction, with the reflected rays moving together while maintaining consistency. Diffuse reflection occurs when light hits a rough, irregular surface and is scattered in various directions. Often, in everyday life, light reflection combines specular and diffuse reflection to generate white light. Light reflection may obscure an important part of the lesion on the screen, thereby interfering with clear monitoring.

Unpolarized light vibrates in various directions and becomes polarized when it passes through a PL filter. Polarization refers to a form of light energy in which magnetic or electric fields make up wave vibration in a specific direction. If the polarized light is guided to pass through the PL filter again, the angle between the direction of vibration of the polarized light and the direction of the PL filter is defined as *θ*. When polarization is defined as *H_p_*, the light passing through the PL filter is defined as H_p_ cos*θ* [19,22].

When an image is captured with a camera, the energy of the image exists in vertical polarization (*V_in_*) and horizontal polarization (*H_in_*) [19,20]. If an image is captured using a camera without a PL filter, both vertical polarization (*V_in_*) and horizontal polarization (*H_in_*) are incident on the camera, as shown in Figure 6a [20,26].

The image will have the diffuse reflection characteristics of the white light source, as shown in Figure 6b. Therefore, it is necessary to remove the vertical polarization (*V_in_*), leaving only the horizontal polarization (*H_in_*). To remove the vertical polarization (*V_in_*), the F_2_ filter is rotated while keeping the F_1_ filter fixed, as shown in Figure 7 [20,26]. At this time, when the rotation angle (*θ*) of the F_1_ filter is 0°, the rotation angle (*θ*) of the F_2_ filter can be adjusted from 0° to 360°.

If the F_2_ filter rotates (p_2_) while the direction angle (p_1_) of the F_1_ filter is 0°, P_3_ obtains a polarization effect, as given by Equation (2), and begins to reduce light reflection [20,26]. For example, when the intensity of light reflection (*I_ref_*) is 0° for the F_1_ filter and 90° for the F_2_ filter, the phase angle (*θ*) of the intensity of light reflection passing through the F_2_ filter will be 90° (*θ* = 90°), which effectively eliminates light reflection. Therefore, if *I_Ep_* is 50 mW/cm^2^ (*θ* = 0°), as shown in Table 1, *I_ref_* will be 0 mW/cm^2^ (*θ* = 90°).
(2)$$I_{ref}=I_{EP}{cos}^{2}\theta_{EP}$$

Consequently, when the intensity of light reflection decreased, as shown in the sine waveform in Figure 8, and the rotation angle (*r_2_*) of the F_2_ filter was set to 90° or 270°, the intensity of light reflection became 0 mW/cm^2^, and the light reflection was eliminated. 

## 3. Experimental Results

### 3.1. Experimental Environment Configuration

A phantom was manufactured, as shown in Figure 9, to obtain the experimental results through fluorescence emission. The phantom model had various curves that provided the optimal conditions for inducing light reflection.

To prepare the phantom, as shown in Figure 10, 3 cc of fluorescein sodium 10% injection solution and 97 cc of sterile distilled water were mixed, and the mixture was combined with 20 mL of silicone rubber solution. The resulting mixture was injected into a silver cell dish and then heated to a temperature of 60 °C for approximately 6 h to produce a solid phantom. The size of the phantom was designed to be similar to that of a specimen extracted from the body. The phantom, which had various curves and moisture contents, provided an environment conducive to light reflection.

After constructing a square box using 3D printing technology, four mirrors were installed inside the square box to increase the beam width and beam intensity, as shown in Figure 11, replicating the setup used in a previous study [9]. However, this structure requires the insertion of a polarizing lens to eliminate light reflection; therefore, it was modified to suit the experimental environment. When LED light is incident, the beam width expands through refraction and reflection via mirrors, thereby minimizing the beam intensity loss and enhancing the beam strength [9].

The rectangular box was designed to accommodate an additional LED and a near–infrared (NIR) camera. A lens capable of achieving a polarization filtering effect was placed above the mirror facing the LED. The lens is rotatable, allowing for the physical adjustment of the light reflection depending on the angle of rotation. As shown in Figure 12, the lens was configured to be detachable/attachable, enabling the observation of differences in light reflection depending on whether the lens was present. To block external light, a square box was constructed with black cases to absorb unnecessary light and strengthen the shielding effect. The height of the mirror from the ground was 31.55 mm, which was determined to be optimal for increasing the beam width (20.4° → 40.5°) and power [9]. The floor included a dedicated phantom capable of fluorescence emission, graph paper, and power sensor.

As shown in Figure 13, the phantom can be used to examine the occurrence and reduction of light reflection. As it is mixed with a fluorescent contrast agent, fluorescence emission is possible, making it easy to observe fluorescence color images. Four mirrors, LEDs, and a camera were installed inside the square box, followed by a rotatable lens on the LED surface to control the polarization direction. An experiment was conducted as shown in Figure 13. The four mirrors were placed inside the box to increase the beam width and beam intensity.

### 3.2. Experiment Results

The results of increasing the beam width, equalizing the beam power intensity, and minimizing the power loss are shown in Figure 14. To obtain these quantitative results, we measured the beam irradiation range and power on the graph paper using a ruler and power sensor. In this experiment, the wavelength band of the LED was 405 nm and the WD was 18 cm. Each square on the graph paper had a side of 1 cm. Therefore, we arbitrarily assigned a code from 0 to *f* to each coordinate and used the ruler and sensor to measure the beam width and power at each coordinate position.

Without a beam mirror, the beam had an irradiation radius of 4.4 cm, and the beam power was 13.6 mW. However, when a beam mirror was installed, the beam width increased without a loss of beam power. When the LED beam was irradiated without a beam mirror (without a PL filter), the beam width spread to an area of 20.4° (diameter: 4.4 cm, from 0 to >b). The beam power remained at 13.6 mW. However, at coordinate “”., the beam power rapidly decreased to 0.73 mW, and when the beam moved past position “”., the power approached zero. When the LED was irradiated with the beam mirror mounted, the beam width expanded to 40.5° (diameter: 6.5 cm, from 0 to >c), increasing the beam width while minimizing the beam power loss. Therefore, on the graph paper, the power at coordinate 0 was still measured to be 13.6 mW; however, as the beam intensity decreased with increasing distance, the beam power was measured as 12.1 mW at position “”. and 10.9 mW at position “d”. Beyond this position, the power gradually decreased.

During the experiment, the LED beam wavelength was directed at the fluorescence emission phantom, and the emission wavelength from the phantom was detected using a camera sensor, as shown in Figure 15, with the fluorescence color observed on an external monitor. The camera was equipped with a long–pass filter to allow only the fluorescence emission wavelength to pass. Regarding the two rotatable lenses that enabled the adjustment of the PL filter attached to the LED surface, one lens was fixed at 0°, whereas the other could rotate between 0° and 360° to position the two filters at an angle of 90°.

When the lens angle was adjusted to 0°, as shown in Figure 16, 100% light reflection occurred, and the entire surface of the phantom appeared white. When the lens was rotated at intervals of 10°, the light reflection gradually decreased by 80% as the angle approached 80°. When the lens was rotated to 90°, more than 98% of the light reflection was eliminated. When the lens was rotated to 260°, light reflection reappeared, increasing by approximately 30%. At 270°, more than 98% of the light reflection was removed, confirming that light reflection was most effectively eliminated at 90° and 270°.

Figure 17 shows the experimental process for obtaining the light reflection results using a phantom that was melted and remade to resemble the size of a lesion.

As shown in Figure 18, LED irradiation experiments were conducted to determine whether the beam width covered the entire phantom and provided sufficient power for the fluorescence emission. Satisfactory results were obtained by measuring the LED beam width and emission power during light reflection. The LED beam width was sufficient to cover the entire phantom. Both the beam width and beam intensity increased, enabling sufficient fluorescence emission. In addition, light reflection was almost completely eliminated, yielding results that can be applied in clinical settings.

For the LEDs without a mirror, the fluorescence emission range was observed to be 3.25 cm (from 0 to a) in diameter in the brightest state. The point at which the fluorescence emission ended (from a to c) was 3.32 cm in diameter. At point (c), the fluorescence emission intensity was weak. Therefore, standard LEDs can only observe tumors with an area of approximately 3.0 cm in diameter. However, with the mirror–inserted LED, the beam width increased by more than 2.0 times, and the fluorescence emission was observed in its brightest form across a diameter of approximately 6.5 cm (from a to d), with uniform intensity across a certain area. In addition, the region with excellent fluorescence emission (from 0 to c) had a diameter of approximately 6.5 cm.

## 4. Discussion

As the proposed method combines a beam mirror and PL filter, it can be used with general LEDs, regardless of the wavelength and type.

The physical and chemical properties of fluorescent contrast agents are determined by the size, composition, and wavelength of light used for illumination. Organic fluorescent contrast agents are typically composed of small molecules, which, when injected into the body, have a relatively short half−life, making them useful for detecting short−term biological processes [28]. Notable examples of organic fluorescent contrast agents are indocyanine green, fluorescein sodium, and rhodamine. In particular, NIR fluorescent contrast agents, such as indocyanine green and fluorescein sodium, can be measured using high−sensitivity video cameras at NIR wavelengths. By utilizing optical techniques in the NIR range after injecting a fluorescent contrast agent into the tissue, it is possible to minimize the effect of autofluorescence caused by proteins or small molecules in the tissue and observe relatively deep tissues. Therefore, NIR agents with low biological background signals are preferable when selecting a fluorescent contrast agent. It is believed that a separate circuit configuration and a new semiconductor process are necessary to maintain the beam intensity, increase the beam width, and eliminate light reflection. However, the proposed method can achieve effects similar to those of semiconductor processes by combining the performances of commercially available LEDs. Therefore, if the proposed method is applied each time the LED is changed, it is believed that both increasing the beam width and beam intensity and reducing light reflection will be possible. According to the Malus law, a PL filter must be installed on the camera to eliminate light reflection. However, in the proposed method, a PL filter was installed on the LEDs.

The proposed system successfully reduced light reflection using a linear PL filter. Owing to this reduction in light reflection, the field of view for observing lesions was secured, and clearer images were obtained. However, when a linear PL filter is inserted, the image may appear dark owing to the resistance of the lens material [29]. In this case, increasing the light−source intensity can result in a brighter image [29].

The methods in [9] and the proposed method differ in beam intensity and beam width owing to the differences in the WD. However, if the WD is the same, results with similar light intensities and beam widths can be obtained. As the WD varies depending on the clinical site, we determined the appropriate WD value for the surgical diagnosis environment during testing. When the rotation angle of the PL filter was 90°, the image became slightly darker; however, the observation was still possible. Further investigation is required to address this darkening effect.

The advantage of observing lesions by inducing fluorescence emission through irradiation with an external light source after the injection of a fluorescent contrast agent is that blood vessels (or surrounding tissues) and tumors can be clearly distinguished by color [30]. A wavelength band of 405 nm was used for the light source irradiation. Blue light at 405 nm is commonly used in medical sensors, diagnostics, and sterilization [31]. This is because prolonged exposure to this wavelength can cause damage to the human skin or eyes, requiring special precautions. However, as this study only aimed to determine the presence of residual tumors, safety can be assured because the light source is irradiated for a duration shorter than 1 min.

Fluorescent contrast agents may temporarily cause yellow urine excretion; however, this usually disappears within 24–48 h [30].

When designing the proposed structure, external wavelength bands, such as those from fluorescent lamps and lighting, can be transmitted. If these wavelengths are mixed with the LED wavelength, they can interfere with fluorescence emission. Therefore, to block the transmission of the external wavelengths, a UV bond was used to seal the gaps or edges of the structure, as shown in Figure 19. In addition, when manufacturing the outer structure via 3D printing, a black material is used to suppress the transmission of external wavelengths and ensure that they are absorbed by the outer structure. Moreover, when the LED light source and fluorescence emission wavelength are mixed and captured by the camera, the monitor image may become distorted, making it difficult to observe the lesion. To prevent this, a filter was installed on the camera head to block the LED wavelength, allowing only the fluorescence emission wavelength to be transmitted to the camera.

As the proposed method was tested using a phantom, its clinical suitability needs to be verified through future animal experiments. Therefore, enhancing image brightness and conducting animal testing are planned for future studies. The reductions in light reflection are compared in Table 2, where the differentiation and superiority of the proposed method are presented [32,33,34,35].

In cases where the WD is short, even if the beam power of the LED is relatively low, the loss of power reaching the target will not be significant. In particular, the power delivered by lasers to the target is higher than that delivered by the LEDs [35]. Even a single laser can produce high power. However, lasers generate significant heat, which leads to high power consumption and severe thermal destruction of the laser module. Additionally, owing to their potential harm to the human body, lasers impose a greater burden on the medical device licensing process. However, note that a higher LED power is not the only factor that determines performance [32,33,34,35]. It is crucial to have sufficient power to enable fluorescence emission; however, the wider the beam width, the better the performance. This is because a wider beam can irradiate the entire lesion, enabling sufficient fluorescence emission from the entire lesion. Therefore, LEDs with wide beam widths are considered to exhibit an excellent performance.

## 5. Conclusions

Diagnostic endoscopes or microscopes used for observing fluorescence emission often have a narrow field of view or limited observation area when examining lesions owing to the limited beam width of the LED. This limitation causes inconvenience because it requires frequent adjustments to the camera position and changes in the direction angle. Medical diagnostic equipment and cameras are often large and heavy, making directional changes difficult.

This paper proposed a method to effectively eliminate light reflection while increasing the beam intensity and beam width using a beam mirror. The beam intensity and beam width could be increased by equipping the LED with a beam mirror. However, an even more outstanding feature is that light reflection can be effectively removed by applying a PL filter, and the increased beam width allows the observation of wider lesions through fluorescence emission.

As a PL filter is combined with an LED, the loss of image quality from the camera can be minimized. Although these results were obtained using an experimental setup, we believe that this method will be useful in clinical settings if the system is miniaturized and manufactured using precise 3D printing technology. The proposed method can be applied in the field of fluorescence emission diagnostics and may also be useful in lighting systems for endoscopic diagnostic imaging or for observing lesion specimens. Therefore, focusing on procedures and diagnoses is expected to be particularly beneficial for surgical and other specialties.

## Figures and Tables

**Figure 1 diagnostics-14-01996-f001:**
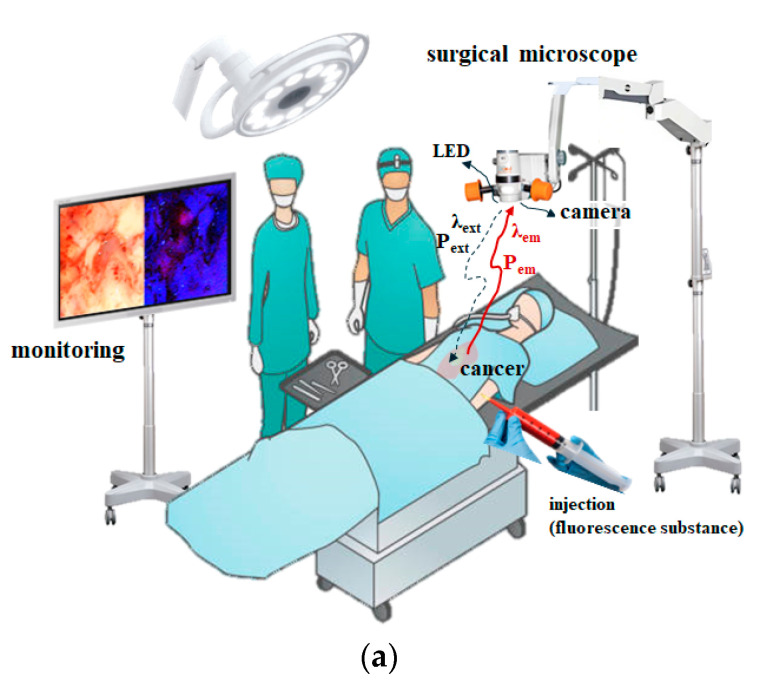
Fluorescence emission diagnosis using a surgical microscope; (**a**) surgical microscope monitoring; (**b**) fluorescence emission diagnosis [20]; (**c**) surgical incision and fluorescent contrast medium observation procedure.

**Figure 2 diagnostics-14-01996-f002:**
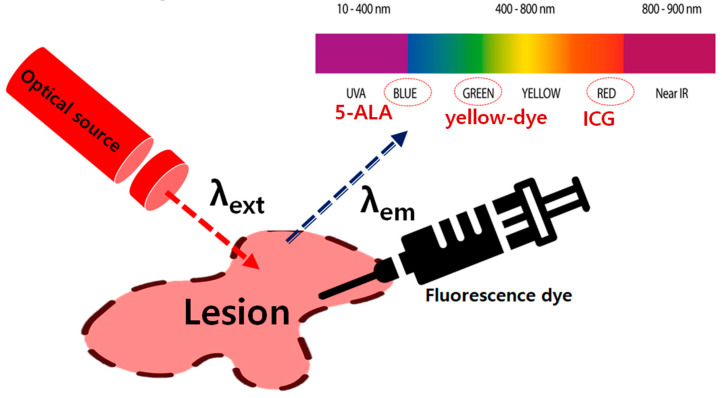
Optical source irradiation with excitation wavelength and the expression of color imaging with fluorescence emission wavelength.

**Figure 3 diagnostics-14-01996-f003:**
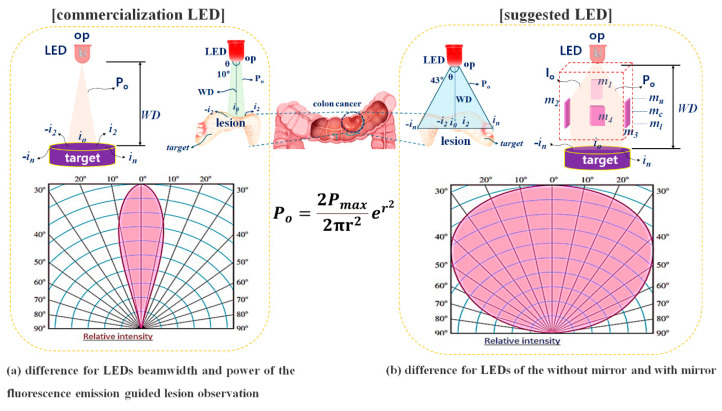
Difference in fluorescence emission lesion observation area according to LED beam width.

**Figure 4 diagnostics-14-01996-f004:**
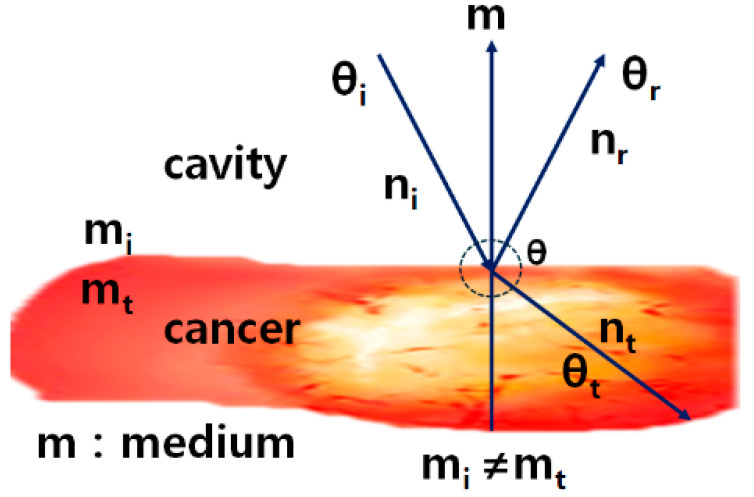
Analysis of light reflection generation using Snell’s law.

**Figure 5 diagnostics-14-01996-f005:**
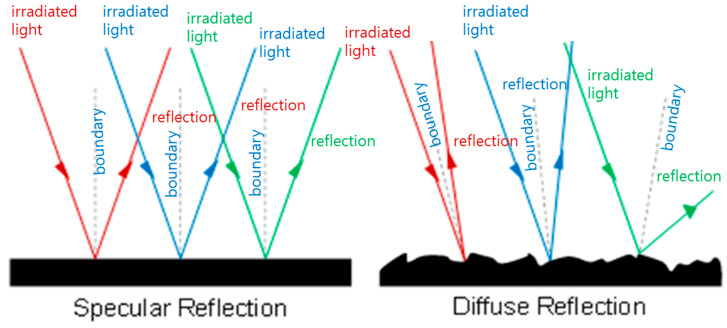
Analysis of light reflection and linear polarization.

**Figure 6 diagnostics-14-01996-f006:**
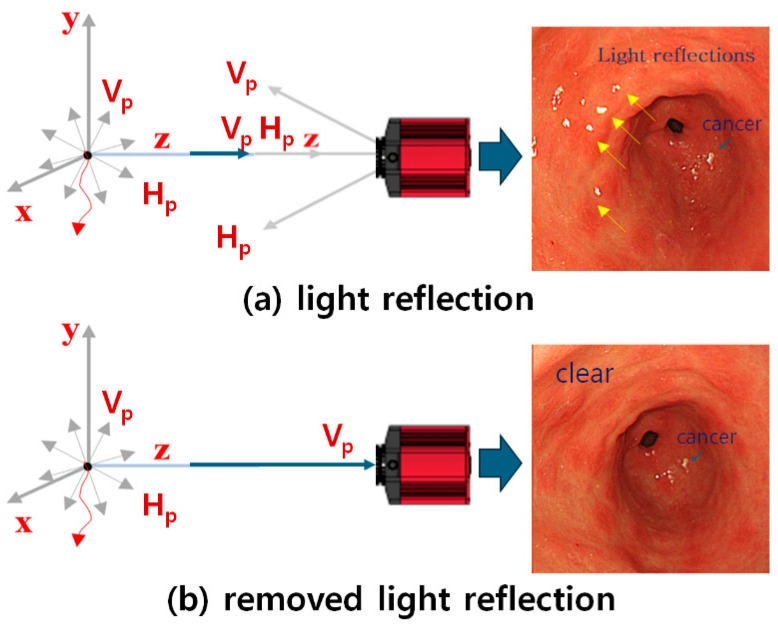
Light reflection phenomenon during the camera shooting process.

**Figure 7 diagnostics-14-01996-f007:**
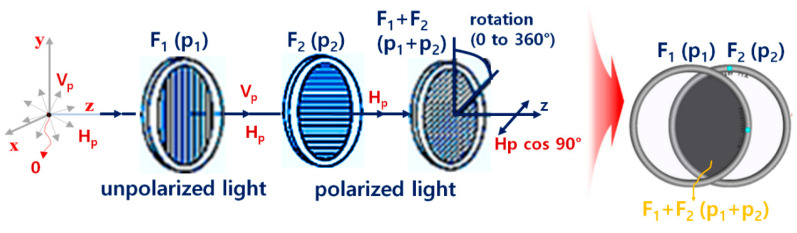
Analysis of the light reflection removal method.

**Figure 8 diagnostics-14-01996-f008:**
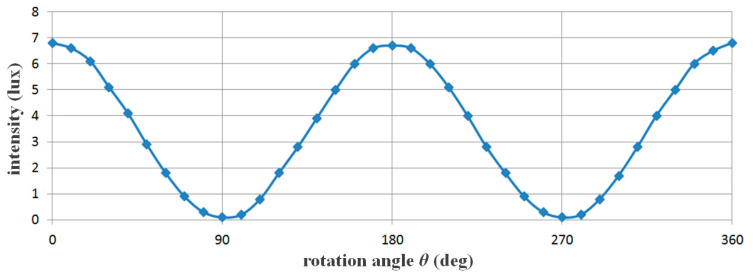
Changes in light reflection intensity according to the rotation axis of the filter using the Malus law.

**Figure 9 diagnostics-14-01996-f009:**
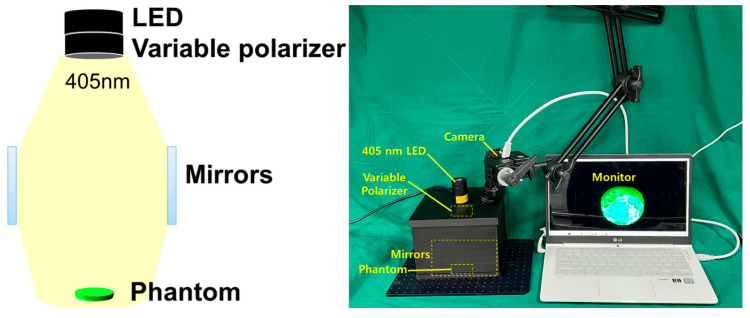
Experimental environment configuration.

**Figure 10 diagnostics-14-01996-f010:**
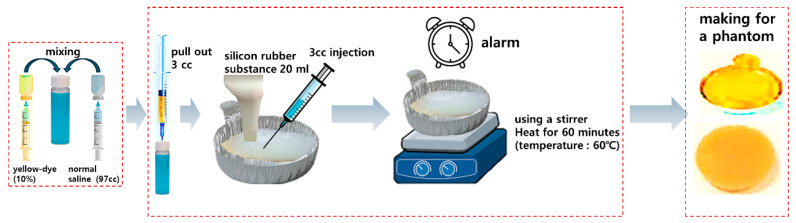
Fluorescent sodium (yellow dye) phantom production process.

**Figure 11 diagnostics-14-01996-f011:**
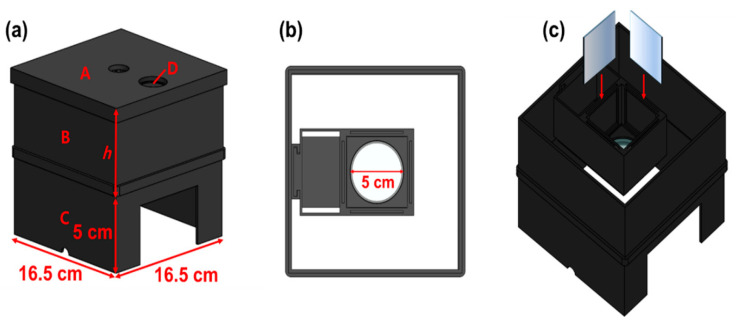
Production of the experimental environment structure: (**a**) external, (**b**) top, and (**c**) internal (A: cover area, B: beam mirror shield cover area, and C: light source irradiation shield cover area).

**Figure 12 diagnostics-14-01996-f012:**
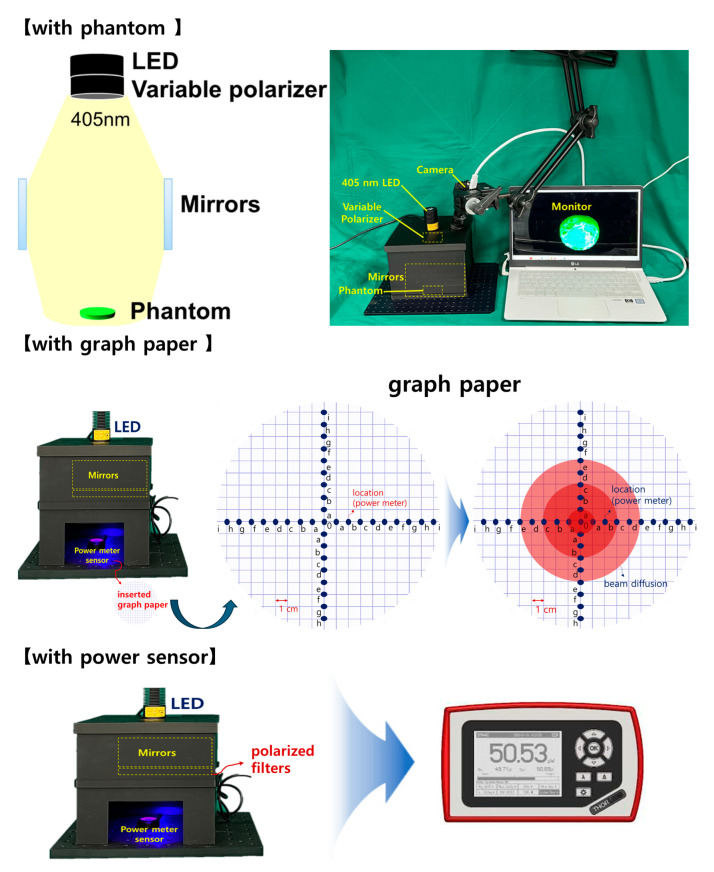
Experiment sequence and result derivation method.

**Figure 13 diagnostics-14-01996-f013:**
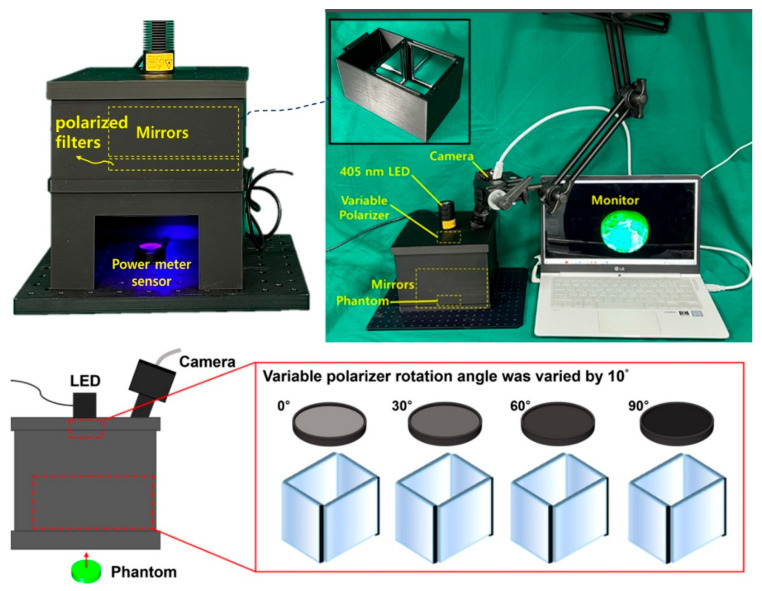
Structural design process for beam width expansion and reflection removal.

**Figure 14 diagnostics-14-01996-f014:**
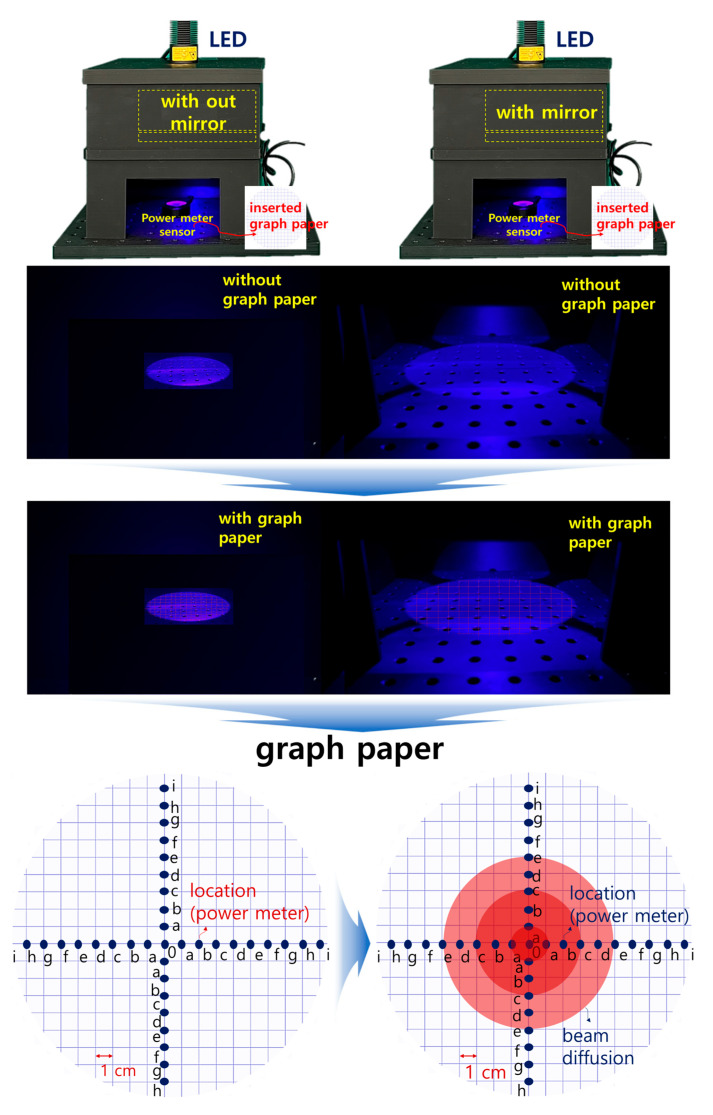
Methods for achieving increased beam intensity and beam width results.

**Figure 15 diagnostics-14-01996-f015:**
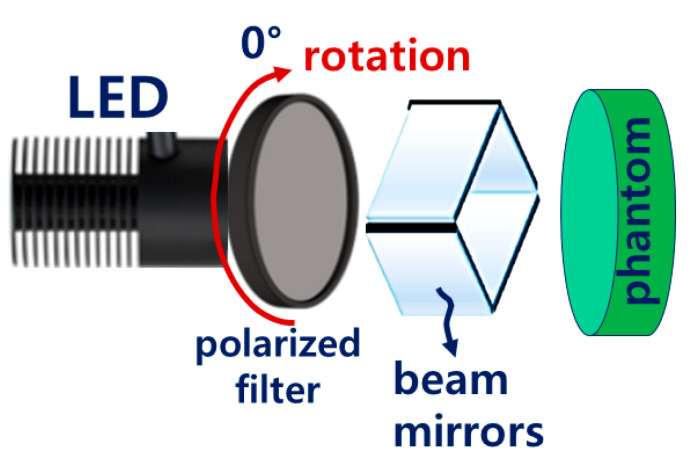
LED connection process (PL filter and beam mirror), and phantom experiment form.

**Figure 16 diagnostics-14-01996-f016:**
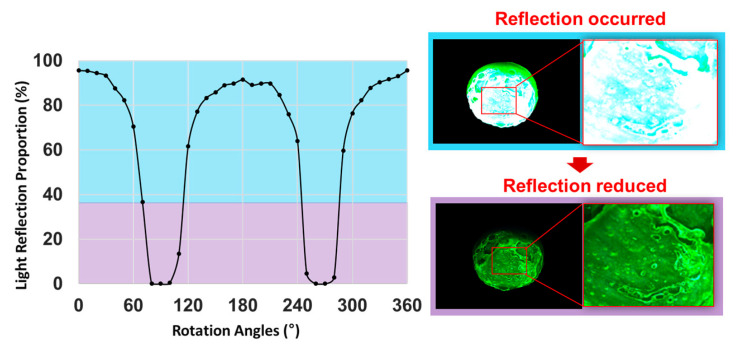
Phantom monitoring result with the variation of the polarization angle.

**Figure 17 diagnostics-14-01996-f017:**
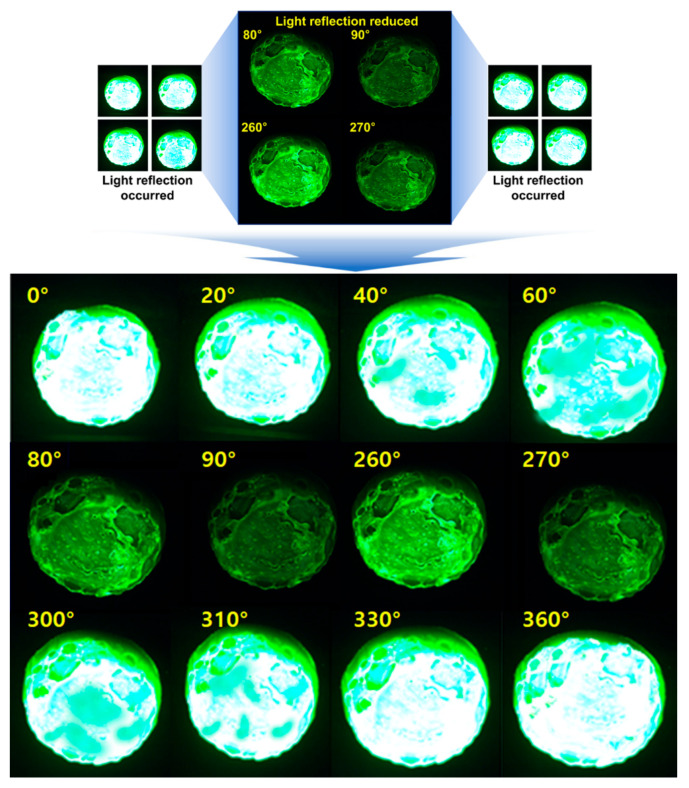
Results of the light reflection removal experiment.

**Figure 18 diagnostics-14-01996-f018:**
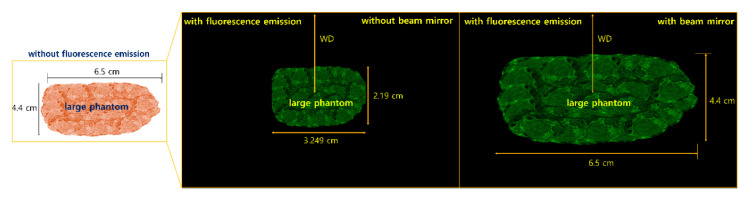
Results of the fluorescence emission experiment on the entire surface of the phantom following LED beam irradiation on a phantom similar to the size of the lesion (beam width, removal of light reflection, and the possibility of fluorescence emission on the entire surface of the phantom).

**Figure 19 diagnostics-14-01996-f019:**
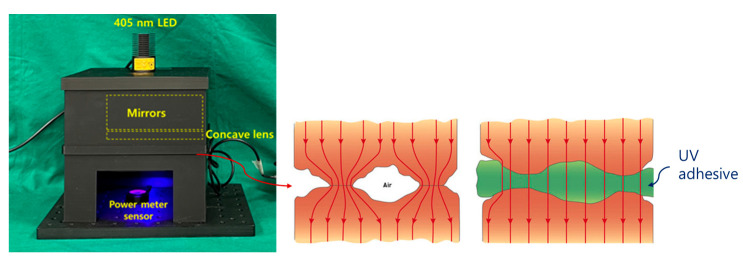
Gap UV adhesive treatment to block external light wavelength interference.

**Table 1 diagnostics-14-01996-t001:** Analysis of light reflection removal using rotation angle information (F_2_).

F_2_ Rotation Angle of Filter (θ)	Light Reflection Intensity [mW/cm^2^]	F_2_ Filter Rotation Angle (θ)	Light Reflection Intensity [mW/cm^2^]
0°	50.0	210°	37.5
30°	37.5	240°	12.5
60°	50.0	270°	0.00
90°	0.00	300°	12.5
120°	12.5	330°	37.5
150°	37.5	360°	50.0
180°	50.0		

**Table 2 diagnostics-14-01996-t002:** Analysis of comparison and difference for the suggested LED and others.

Ref. [#]	λ_ext_ [nm]	WD [cm]	Beam Width [cm^2^]	P_max_ [mW]	Target Received Power [mW]	LED Quantity [ea]
This study	405	18.0	6.50	200	10.9	1.00 (LED)
[32]	550	56.75	5.72	7920	0.196	9.00 (LED)
[33]	625	30.00	3.24	300,300	26.55	130 (LED)
[34]	467	6.17	3.31	100	6.10	52 (LED)
[35]	405	0.25	0.027	40.0	12.3	1.0 (laser)

## Data Availability

The data presented in this study are available upon request from the corresponding author. The data are not publicly available due to privacy and ethical restrictions.

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
