# Peer review of "Optical Methods for Optimizing Fluorescence Imaging Field of View and Image Quality in Surgical Guidance Procedures"

_diagnostics, 2024, doi:10.3390/diagnostics14171996_

Round 1
Reviewer 1 Report
Comments and Suggestions for Authors
In this paper, the authors propose a method of using beam mirrors to effectively eliminate light reflections when increasing beam intensity and beam width. The authors highlight the advantages of this method and also perform relevant validations. Why don't the authors apply this method to small animal in vivo tumour studies? And what about the background fluorescence of the living tissue itself?
Comments on the Quality of English Language
Too many English phrases and many punctuation errors
Round 1
Comments 1
n this paper, the authors propose a method of using beam mirrors to effectively eliminate light reflections when increasing beam intensity and beam width. The authors highlight the advantages of this method and also perform relevant validations. Why don't the authors apply this method to small animal in vivo tumour studies? And what about the background fluorescence of the living tissue itself?.
Answer 1
We focused on the experiment on the possibility of increasing the beam width and intensity of the single LED. To obtain these results, we analyzed them from a physical perspective and submitted a paper, which is currently under evaluation. We plan to discover new research topics and conduct animal experiments this winter. If we conduct experiments on living tissues themselves through preclinical testing, tumors will appear yellow or light green due to the fluorescent contrast agent, and other tissues will appear dark. As shown in Figure 18, if we irradiate the LED on tissues with tumors attached, light will be irradiated over a fairly large area. In addition, if we conduct monitoring by inducing fluorescence emission, we expect that we will be able to observe a large area of ​​tissue.
Reviewer 2 Report
Comments and Suggestions for Authors
While appreciating your efforts, I have identified several areas that could benefit from improvement:
· The introduction requires further elaboration to provide a more comprehensive context for the study.
· Although the article focuses on fluorescence imaging, there is no detailed introduction of fluorescence imaging itself. Consider including a background on fluorescence imaging, its principles, and its significance in surgical procedures.
· The second paragraph in Section 2.1 is poorly written and needs major revision.
· It would be beneficial to add information about the fluorophores currently used for fluorescence-guided surgery, their emission wavelengths, and the associated problems based on the wavelength of emission.
· Additionally, the paper should discuss the problems and advantages of different regions of the spectrum (visible vs. near-infrared).
· There is a typographical error in line 104 that needs correction. Please rewrite the paragraph for clarity and accuracy.
I believe addressing these points will significantly enhance the quality and impact of your manuscript. I appreciate your dedication to advancing scientific knowledge.
Round 2
Comments 1
While appreciating your efforts, I have identified several areas that could benefit from improvement:
The introduction requires further elaboration to provide a more comprehensive context for the study..
Answer 1
Thank you very much for your review of my manuscript. I have tried my best to respond to your comments. Please review. Thank you.
Comments 2
Although the article focuses on fluorescence imaging, there is no detailed introduction of fluorescence imaging itself. Consider including a background on fluorescence imaging, its principles, and its significance in surgical procedures.
Answer 2
I marked lines 142-160 (pink) at the bottom of Figure 1. I included Figure 1(b), (c) and the explanation for the advice.
Comments 3
The second paragraph in Section 2.1 is poorly written and needs major revision.
Answer 3
I have corrected it. Please refer to lines 126-132 and 161-168 (yellow). Thank you.
Comments 4
It would be beneficial to add information about the fluorophores currently used for fluorescence-guided surgery, their emission wavelengths, and the associated problems based on the wavelength of emission.
Answer 4
I have written about the problems and countermeasures of fluorescent contrast agents. Please refer to lines 441-442 (blue) in the discussion section. I have also written about the problems of LED light source wavelength and fluorescent emission wavelength. I have also written about the relationship between the black appearance design and wavelength. Please refer to lines 443-453 (blue) in the discussion section and Figure 19.
Comments 5
Additionally, the paper should discuss the problems and advantages of different regions of the spectrum (visible vs. near-infrared).
Answer 5
I wrote about the problems in the 405 nm wavelength band. Please check lines 432-440 (red) in the Discussion section.
Comments 6
There is a typographical error in line 104 that needs correction. Please rewrite the paragraph for clarity and accuracy.
Answer 6
Thank you, I've made some overall corrections.
Comments 7
I believe addressing these points will significantly enhance the quality and impact of your manuscript. I appreciate your dedication to advancing scientific knowledge.
Answer 7
Thank you for your advice on improving the quality of the manuscript. Please take good care of it for publication. I will also do my best.
Round 2
Reviewer 1 Report
Comments and Suggestions for Authors
In lines 151-152, the authors describe the formulation of fluorescent contrast agents:”To prepare the solution, 500 mg of fluorescent contrast agent is mixed with 10 mL of normal saline, resulting in a concentration of 50 mg/kg.” Why is such a large concentration of fluorescent contrast agent formulated? Firstly, a large concentration of fluorescent contrast agent will lead to aggregation induced quenching (ACQ). Secondly, such a large concentration of fluorescent contrast agents is theoretically poor biocompatibility.
The author only considers the fluorescence emission of the fluorescent contrast agent from a physical point of view, but there will be a large number of proteins or small molecules in the actual organism that will fluoresce with the fluorescent contrast agent, which will eventually cause false positive results. If the authors focus only on experiments with the possibility of increasing the beam width and intensity of a single LED, then the fluorescent contrast agent for the discussion section should be selected as a near-infrared fluorescent contrast agent (such as ICG) because of the low biological background of near-infrared light.
Round 2
Comments 1
While appreciating your efforts, I have identified several areas that could benefit from improvement:
The introduction requires further elaboration to provide a more comprehensive context for the study..
Answer 1
Thank you very much for your review of my manuscript. I have tried my best to respond to your comments. Please review. Thank you.
Comments 2
Although the article focuses on fluorescence imaging, there is no detailed introduction of fluorescence imaging itself. Consider including a background on fluorescence imaging, its principles, and its significance in surgical procedures.
Answer 2
I marked lines 142-160 (pink) at the bottom of Figure 1. I included Figure 1(b), (c) and the explanation for the advice.
Comments 3
The second paragraph in Section 2.1 is poorly written and needs major revision.
Answer 3
I have corrected it. Please refer to lines 126-132 and 161-168 (yellow). Thank you.
Comments 4
It would be beneficial to add information about the fluorophores currently used for fluorescence-guided surgery, their emission wavelengths, and the associated problems based on the wavelength of emission.
Answer 4
I have written about the problems and countermeasures of fluorescent contrast agents. Please refer to lines 441-442 (blue) in the discussion section. I have also written about the problems of LED light source wavelength and fluorescent emission wavelength. I have also written about the relationship between the black appearance design and wavelength. Please refer to lines 443-453 (blue) in the discussion section and Figure 19.
Comments 5
Additionally, the paper should discuss the problems and advantages of different regions of the spectrum (visible vs. near-infrared).
Answer 5
I wrote about the problems in the 405 nm wavelength band. Please check lines 432-440 (red) in the Discussion section.
Comments 6
There is a typographical error in line 104 that needs correction. Please rewrite the paragraph for clarity and accuracy.
Answer 6
Thank you, I've made some overall corrections.
Comments 1
In lines 151-152, the authors describe the formulation of fluorescent contrast agents:”To prepare the solution, 500 mg of fluorescent contrast agent is mixed with 10 mL of normal saline, resulting in a concentration of 50 mg/kg.” Why is such a large concentration of fluorescent contrast agent formulated? Firstly, a large concentration of fluorescent contrast agent will lead to aggregation induced quenching (ACQ). Secondly, such a large concentration of fluorescent contrast agents is theoretically poor biocompatibility.
Answer 7
Your opinion is correct. It was my mistake. The mixing ratio of saline solution was written incorrectly. I modified the 150-160 line (yellow) to reflect your opinion.
The method of preparing fluorescent contrast agent is important for lesion monitoring. Prepare a 5 mg/mL dilution by mixing 500 mg of fluorescent contrast agent with 100 mL of normal saline. Prepare a vial phantom in a 1 mL microfuge tube with 0.8 mL of this mixture. The injection volume is maintained at 0.2–0.4 mL [21].
[21] Lee, S., Yoon, K., Kim, J., & Kim, K. G. (2022). Specular reflection suppression through the adjustment of linear polarization for tumor diagnosis using fluorescein sodium. Sensors (Basel, Switzerland), 22(17), 6651.
After injection of fluorescent contrast agent, the uptaken tumor appears yellow or light green, and the wavelength band of 530~560nm can be confirmed through spectrum measurement. As shown in Figure 1(c), the removal status is observed after the tumor is removed during laparotomy. After injection of fluorescent contrast agent, a picture is taken with a camera and the LED is prepared for irradiation. At this time, the operating room lights are turned off to maintain darkness, and the color expression is observed to monitor the tumor removal status.
Comments 2
The author only considers the fluorescence emission of the fluorescent contrast agent from a physical point of view, but there will be a large number of proteins or small molecules in the actual organism that will fluoresce with the fluorescent contrast agent, which will eventually cause false positive results. If the authors focus only on experiments with the possibility of increasing the beam width and intensity of a single LED, then the fluorescent contrast agent for the discussion section should be selected as a near-infrared fluorescent contrast agent (such as ICG) because of the low biological background of near-infrared light.
Answer 7
[Insert into Discussion]
Please refer to the yellow description on lines 411-423.
The physical and chemical properties of fluorescent contrast agents are determined by the size and composition of the contrast agent, and the wavelength of light used for illumination. Organic fluorescent contrast agents are generally composed of small molecules, and when injected into the body, they have a relatively short half-life, making them useful for detecting short-term biological processes [28].
[28] Ewelt C, Nemes A, Senner V, et al.: “Fluorescence in neurosurgery: Its diagnostic and therapeutic use. Review of the literature” J. Photochem. Photobiol. B, 2015, 148, pp. 302–309.
Representative examples of organic fluorescent contrast agents include indocyanine green, sodium fluorescein, and rhodamine. In particular, indocyanine green and sodium fluorescein, which are near-infrared (NIR) fluorescent contrast agents, can measure fluorescence using a high-sensitivity video camera in the NIR wavelength. When using optical techniques in the near-infrared wavelength after injecting a fluorescent contrast agent into the tissue, there is an advantage in that the influence of autofluorescence caused by proteins or small molecules in the tissue can be minimized and relatively deep tissues can be observed. For this reason, when selecting a fluorescent contrast agent, it is preferable to use a near-infrared fluorescent contrast agent with low biological background signals.
Round 3
Reviewer 1 Report
Comments and Suggestions for Authors
As the authots done perfectly modification work, I am pleased to accept it for the publication.